# An examination of early socioeconomic status and neighborhood disadvantage as independent predictors of antisocial behavior: A longitudinal adoption study

**Shelley A. Gresko** [1,2]*, **Laura K. Hink**[1,2], **Robin P. Corley**[2], **Chandra A. Reynolds**[1,2,3], **Elizabeth Muñoz**[4], **Soo Hyun Rhee**[1,2]

1 Department of Psychology and Neuroscience, University of Colorado Boulder, Boulder, Colorado, United States of America, 2 Institute for Behavioral Genetics, University of Colorado Boulder, Boulder, Colorado, United States of America, 3 Department of Psychology, University of California Riverside, Riverside, California, United States of America, 4 Department of Human Development and Family Sciences, University of Texas at Austin, Austin, Texas, United States of America

* shelley.gresko@colorado.edu

**Data Availability Statement:** Given that the data contain potentially identifying and sensitive patient information, data cannot be shared publicly

## Abstract

The present study examined early socioeconomic status (SES) and neighborhood disadvantage (ND) as independent predictors of antisocial behavior (ASB) and addressed the etiology of the associations (i.e., genes versus the environment) using a longitudinal adoption design. Prospective data from the Colorado Adoption Project (435 adoptees, 598 nonadopted children, 526 biological grandparents of adoptees, 481 adoptive parents, and 617 nonadoptive parents including biological parents of unrelated siblings of adoptees) were examined. SES and ND were assessed during infancy and ASB was evaluated from ages four through 16 using parent and teacher report. Associations between predictors and ASB were compared across adoptive and nonadoptive families and sex. Early SES was a nominally significant, independent predictor of antisocial ASB, such that lower SES predicted higher levels of ASB in nonadoptive families only. ND was not associated with ASB. Associations were consistent across aggression and delinquency, and neither SES nor ND was associated with change in ASB over time. Nominally significant associations did not remain significant after controlling for multiple testing. As such, despite nonsignificant differences in associations across sex or adoptive status, we were unable to make definitive conclusions regarding the genetic versus environmental etiology of or sex differences in the influence of SES and ND on ASB. Despite inconclusive findings, in nonadoptees, results were consistent—in effect size and direction—with previous studies in the literature indicating that lower SES is associated with increased risk for ASB.

## Introduction

Antisocial behavior (ASB) lies on a continuum and encompasses several psychological diagnoses and characteristics including oppositional defiant disorder, conduct disorder, antisocial

because of restrictions put in place by the University of Colorado Boulder Institutional Review Board. Unidentifiable patient data are in the process of being added to the Harvard Dataverse repository. A portion of this data is currently available through the Harvard Dataverse at https://doi.org/doi:10.7910/DVN/BCDSEU (Plomin et al., 2022) and access can be requested by contacting the Manager of Operations for the Henry A. Murray Research Archive, Institute for Quantitative Social Sciences, 1737 Cambridge St, Cambridge, MA 02138, USA., mra@help.hmdc.harvard.edu. Researchers who meet the criteria for access to confidential data may apply for access to data not yet added to the Harvard Dataverse by contacting the corresponding author (shelley.gresko@colorado.edu) and the University of Colorado Boulder Institutional Review Board (irbadmin@colorado.edu). Plomin, R., DeFries, J. C., & Fulker, D. W. (2022). Colorado Adoption Project, 1976-1989 Version V1) [longitudinal, field study, hereditary]. Harvard Dataverse. https://doi.org/doi:10.7910/DVN/BCDSEU.

**Funding:** This work was supported by National Institute on Drug Abuse grants: DA054087, DA011015, DA017637; National Institute on Aging grant: AG046938; and Eunice Kennedy Shriver National Institute of Child Health and Human Development grant: HD010333 SHR received grant #DA054087 from the National Institute on Drug Abuse, URL: https://nida.nih.gov/ John K. Hewitt received grant #DA011015 from the National Institute on Drug Abuse, URL: https://nida.nih.gov/ Jerry A. Stitzel received grant #DA017637 from the National Institute on Drug Abuse, URL: https://nida.nih.gov/ CAR and Sally J. Wadsworth received grant #AG046938 from the National Institute on Aging, URL: https://www.nia.nih.gov/ Sally J. Wadsworth received grant #HD010333 from the Eunice Kennedy Shriver National Institute of Child Health and Human Development, URL: https://www.nichd.nih.gov/ The funders had no role in study design, data collection and analysis, decision to publish, or preparation of the manuscript.

**Competing interests:** The authors have declared that competing interests exist.

personality disorder, and externalizing problems. These behaviors have far reaching negative implications for society and the individual, including crime [1], risk for high school dropout [2], early substance use initiation [3], other psychiatric disorders [4, 5], and premature mortality [5–7], and additional research clarifying the etiology of ASB is needed.

Overall, twin and adoption studies demonstrate that approximately 50% of the variance of ASB is explained by genetic influences, but that ASB is more significantly influenced by the shared environment compared to other behavioral traits [8, 9], with shared environmental influences explaining approximately 20% of the variance [10, 11]. Comprehensive research on the putative environmental influences on ASB is needed to prevent and mitigate ASB effectively.

## Evidence for putative environmental predictors of ASB: Parental socioeconomic status and neighborhood disadvantage

Many studies have investigated parental socioeconomic status (SES) and neighborhood disadvantage (ND) as putative environmental influences on childhood ASB. Two meta-analyses examining the associations between parents' SES and their children's ASB have reported statistically significant, albeit small effects; for example, one review reported a Fisher's $Z = -0.099$, $p < .001$ [12], and another reported that low SES was associated with a 0.28 standard deviation increase in child externalizing behavior [13]. In the first meta-analysis, the sample's SES variance did not significantly moderate associations between SES and ASB. Several studies and one meta-analysis [14] also reported a positive association between higher ND (e.g., neighborhood-level poverty, social housing levels, and unemployment) and various ASB constructs [15–24], beginning as early as toddlerhood [25], although a threshold of high ND may need to be surpassed before children are at increased risk for ASB, specifically in boys [25]. Hypothesized mechanisms of these associations include behavioral modeling (e.g., disadvantaged environments may provide models for ASB [17] and psychological maladjustment to stressors unique to disadvantaged environments [17, 26].

Associations between SES and ASB could be confounded by ND and vice versa, as ND and SES are significantly correlated and families with low SES likely live in more disadvantaged neighborhoods [22]. However, studies indicate that lower SES is an independent predictor of ASB after controlling for ND [21, 23, 24, 27–29] and that higher ND is an independent predictor after controlling for SES [16, 21, 23, 24, 27, 29–33]. In addition, SES and ND may influence ASB independently in toddlerhood, middle childhood, and adolescence; however, most studies have examined predictors and ASB concurrently [19, 23, 27, 32]. Addressing the longitudinal influence of early SES and ND on later ASB is important: socioeconomic adversity may cause severe and prolonged stress during early childhood, in part due to decreased parental resources and negative impacts on parent-infant attachment [34]. Additionally, this toxic stress may impact brain development and the long-term functioning of the biological stress system (i.e., stress response modulation and behavioral regulation) permanently via increased allostatic load [35–37]. Changes in the functioning of the biological stress system mediate the connection between early adversity and negative psychological and behavioral outcomes, such as the poor self-regulation and violence characteristic of ASB [35, 38].

## Associations between SES and ND and ASB across sex and type of ASB

Overall, ASB is more prevalent in boys than girls, with boys displaying more physical or overt aggression and girls displaying more relational aggression [39–41]. Despite these mean differences in ASB prevalence across sex, associations between SES and ASB [12] and ND and ASB

[21, 42] are consistent across sex [12]. However, few studies have examined sex differences in the independent associations across SES, ND, and ASB.

Also, SES and ND may predict ASB differentially depending on the type of ASB examined. For example, one study found that ND predicted nonaggressive (e.g., delinquency), but not aggressive, behaviors after controlling for familial SES [43]. A meta-analysis found no evidence for differences in associations between SES and ASB across aggression and delinquency [12] and included studies that did not control for ND. As the literature on this topic is sparse, more research is needed to clarify discrepancies in the independent influences of SES and ND on aggression versus delinquency.

## Mechanisms influencing associations between parental SES, ASB, and ND: Genetic versus environmental influences

Two hypotheses explicate the mechanisms by which parental SES and ND may influence ASB. The sociogenic or social causation hypothesis [44] implies environmental mediation: that ASB is caused by the stress and adversity associated with lower SES environments. The social selection or downward "drift" hypothesis [45] implies passive $r$GE; i.e., parental genetic predispositions influence associations between the family environment and adolescent characteristics [46–48]. In this context, ASB would lead to diminished occupational and educational gains over time such that lower SES and higher ND is a byproduct of familial ASB history. These two hypotheses are not mutually exclusive, and both may explain associations between SES and ND and ASB.

Support for environmental mediation is suggested by natural and randomized experimental studies concluding that individuals who receive additional income or who experience a change in SES report fewer ASB symptoms [41, 49–52]. Similarly, a randomized experimental study found support for environmental mediation, such that relocation from high to low poverty neighborhoods for adolescents in low-income families was associated with reduced arrests for violent offenses although the authors did not control for change in family SES [53].

## Utilizing adoption studies to distinguish between passive $r$GE versus environmental mediation

It is often assumed that putatively environmental variables such as SES and ND influence ASB purely via the environment. However, environmental exposures and genetic background may be confounded. Adoption studies are a unique way to account for this confound by addressing the role of passive $r$GE versus environmental mediation. Because adoptive parents share only the familial environment with adoptees, similarities between adoptive parents and adoptees are assumed to have a purely environmental etiology. In contrast, similarities between biological parents and their children may be due to both genetic and environmental influences. If associations between predictors and ASB are only due to environmental mediation, we expect to see comparable associations between predictors and outcomes in adoptive and nonadoptive groups. If the association is due to passive rGE only, we would expect to see associations in only nonadoptive families in the absence of selective placement. If both processes influence associations, we would expect associations to be significant in both groups, but higher in nonadoptive families. Neither evocative $r$GE (where environmental responses are evoked by a child's genetically influenced behavior) nor active $r$GE (where children seek environments aligned with their genetic predispositions) are likely explanations for the influence of ND and parental SES in early childhood [46–48] in adoptive or biological families.

## Present study

The present study addressed the independent influence of early parental SES and ND, assessed in children's infancy, on ASB assessed from early childhood through adolescence, using data from the Colorado Adoption Project [54]. Many studies have examined the influence of SES and ND on concurrent ASB, but very few studies have examined the associations between these predictors measured in early infancy with later ASB, despite the well-documented connections between early adversity, toxic stress, and later maladaptive psychological functioning [55, 56]. First, we aimed to replicate previous findings that SES and ND independently predict ASB [21, 23, 24, 27, 29] and extend these findings to association with SES and ND measured in infancy (Aim 1). We also tested whether environmental mediation, passive *r*GE, or both influence these associations, to clarify a gap in the existing literature, which has yet to define the mechanisms by which early SES and ND influence ASB. Second, we explored whether early SES and ND independently predict change in ASB over time (Aim 2). One study on this topic found that boys in deprived neighborhoods were less likely to exhibit declines in ASB compared to girls [21] and another found that low familial SES was associated with an increase in aggression across childhood [57]; however, overall, prior research on the connection between SES, ND, and ASB trajectories is minimal and warrants further examination. In Aim 3, we examined differences in patterns of associations between predictors and aggression (e.g., physical fighting) versus delinquency (e.g., theft and truancy), as these two aspects of ASB are distinct, although there is not conclusive evidence that these have different predictors and etiology [12, 43, 58, 59]. Aims one to three were exploratory, as previous research on these topics is limited with mixed results. Finally, we examined sex differences in associations between predictors and ASB (Aim 4). We did not expect to see significant etiological differences, given lack of reliable sex differences in the existing literature [12, 41, 42].

## Method

### Sample

Participants were from The Colorado Adoption Project (CAP), an ongoing longitudinal adoption study that began in 1975 at the Institute of Behavioral Genetics in Boulder, Colorado. The full sample includes 245 adoptive families, 245 biological parent dyads of adoptees (although most data was on biological mothers only), and 245 control families that were matched on sex of the adoptive child, age and occupational status of the father, and number of children in the family. The present study includes data from 427 adoptees (from 181 girls, 203 boys; either the initial adoptees recruited from the 245 adoptive families or their adopted siblings), 598 nonadopted children (either biological children within adoptive families or children in nonadoptive families; from 284 girls, 314 boys), 526 biological grandparents of adoptees (from 267 grandmothers, 259 grandfathers, including data from grandparents of adopted siblings of initial probands), 486 adoptive parents (from 241 mothers, 245 fathers), and 617 biological parents of nonadoptees (306 mothers, 311 fathers). Prospective recruitment began on January 1st, 1976, and ended on September 30th, 1987. Biological mothers of adoptees were recruited from two large adoption agencies in Colorado. Their children were placed into their adoptive family homes within one year after birth. Social workers matched adoptees with adoptive families on non-proximity of location and similarity of height between adoptive and biological parents. No additional explicit selective placement practices were followed. Seventy-five percent of adoptive parents recruited for the study agreed to participate. Matched control families were recruited from local hospitals. Sibling enrollment and recruitment ended with the last longitudinally followed sibling. Biological and adoptive parents were assessed with a comprehensive

battery of psychological measures when children were one year old or younger. Adoptees and nonadoptees were assessed approximately annually either through home or lab visits or telephone interviews from age one year to 16 years. The sample is over 90% White and is demographically representative of the Denver Metropolitan area at the time of recruitment [60]. Participants self-reported their race as one of five categories: Alaskan Native/American Indigenous (1.4%), Asian (4.7%), Black (0.6%), White (91.6%), and more than one race (0.9%), or unknown or not reported (0.9%). Probands were, on average, adopted within one month of birth, mitigating potential confounds between genetic and environmental influences. Importantly, adoptive and control parents were more likely to be older and in the upper half of SES and upper two-thirds of ND distributions compared to the general population. This project is approved by the University of Colorado Boulder Office of Research Integrity's Institutional Review Board (protocol number 14–0421). Parents completed written consent and children aged seven and older provided written or verbal assent and children provided written consent at age 16. Only participants who provided consent for researchers to geocode first addresses were included in this study. For a detailed description of the recruitment and assessment protocols, refer to Plomin and DeFries [60] and Rhea et al. [54]. Data were collected beginning in 1976. The socioeconomic status and antisocial behavior data were accessed by the authors on April 27[th], 2021, and the neighborhood disadvantage data were finalized and accessed on January 20[th], 2023 (see Methods section for more details on the neighborhood disadvantage variable). De-identified data and code used in analyses are available upon request and, at the time of publication, data are in the process of being added to the Harvard Dataverse.

## Measures

**Socioeconomic status.** SES variables were collected for biological maternal grandparents, adoptive parents, and biological parents of nonadoptees upon entering the study. For adoptive parents and biological parents of nonadoptees, three assessments of familial SES were used: maternal and paternal education (i.e., number of years of schooling completed) and paternal occupational ratings based on the National Opinion Research Center (NORC) 1970 occupational rating scale [61]. Only paternal occupational status was examined; given that the adoption agencies required that one parent be an at-home parent and given the limited number of mothers working outside of the home when the study was initiated, including occupational ratings for mothers would not have meaningfully added to the assessment of familial SES during infancy.

**Neighborhood disadvantage.** The U.S. Census Bureau's Batch Address Geocoder converted participants' addresses at intake to latitude and longitude coordinates and obtained geoidentifiers (GEOID) with resolution to a Census Block. Addresses not located with the U.S. Census tool were entered into Google Maps to identify coordinates. Census tracts respond to population growth; consequently, they change across decennial surveys. Thus, to retain tract boundaries consistent with the 2010 census, we used the Longitudinal Tract Database (LTDB) [62]. First addresses between 1976 and 1989 were matched to the 1980 census which represented 99.7% of the CAP sample, and the remaining .3% to either the 1990 or 2000 census.

We developed an ND composite for addresses using the following Census tract indicators clustered by decennial year (either 1980 or 1990+2000) using the CAP sample and a parallel sample of twins from the Colorado Longitudinal Twin Study (LTS) that were aligned with the 1980 or 1990 census. Each tract was scored on a scale from 0–9 corresponding to that tract's value on a given indicator: percent with a high school degree or less; percent of female headed families; percent unemployment; percent in poverty; median household income (reverse

scored); percent owner occupied units (reverse scored); median contract rent (reverse scored). ND was the mean of decile scores across the seven indicators.

**Antisocial behavior.** Parent-reported ASB was assessed with Achenbach's Child Behavior Checklist (CBCL) [63] externalizing scale at age 4, 7, and yearly from 9–16 years (frequencies and descriptives in S1 and S2 Tables). Teacher-reported ASB was assessed via the Teacher Report Form (TRF) [64] externalizing scale yearly at ages 7–16 years. The externalizing scales consisted of a sum of subscales for aggression (characterized by the sum score of questions such as "physically attacks people" and "threatens people") and delinquency (captured by the sum score of questions such as "truancy, skips school" and "vandalism"). Because the externalizing scale was significantly skewed, the items were binned into ordinal variables, maximizing the number of categories while avoiding small cell sizes (frequencies in S2 Table). This method of transforming non-normally distributed variables into ordinal categories reduces bias by assuming an underlying continuous normal liability distribution [65].

To examine aggression, delinquency, and overall ASB, we conducted confirmatory factors analyses (CFA). CFA were conducted using MPlus Version 8.4 [66]. We used the $\chi^2$ statistic to assess model fit but due to its sensitivity to sample size, we also examined the root mean square error of approximation (RMSEA) [67] and Bentler's Comparative Fit Indices (CFI) [68]. An RMSEA < .06 and a CFI > .95 are indicative of good model fit [69]. The *p*-value of the *z*-statistic, which is the ratio of the parameter estimate to its standard error, was used to determine statistical significance of individual parameter estimates.

For overall ASB, we created a latent hierarchical factor with loadings on a parent reported ASB and teacher reported ASB factors, which in turn had loadings on composite ASB scores for each year of assessment, with auto-residual correlations to account for higher correlations between items in closer temporal proximity (Fig 1). We used CFA instead of exploratory factor analysis (EFA) to capture a broad construct of ASB across childhood and adolescence. The latent hierarchical ASB factor model fit the data well, $\chi^2(806) = 898.55$, *p* = .01, RMSEA = 0.02,

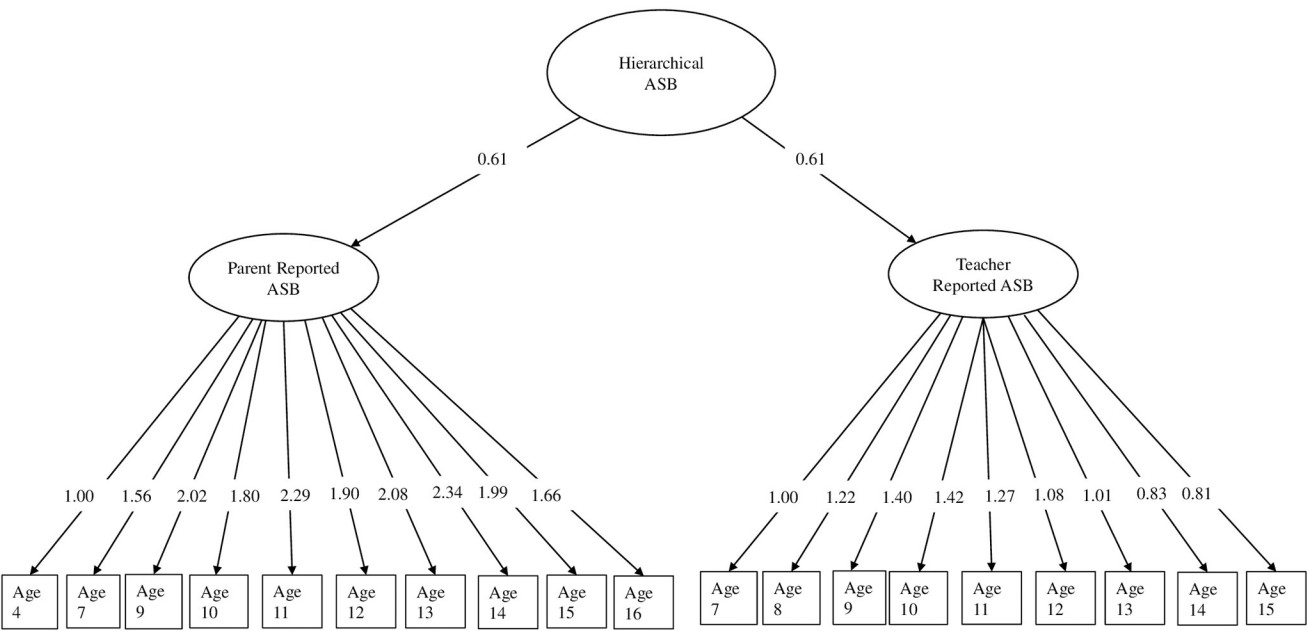

**Fig 1. Hierarchical latent ASB factor.** Note: unstandardized factor loadings presented. All factor loadings significant at p < .001. Model fit: $\chi^2(806) = 898.55$, p = .01, RMSEA = 0.02, CFI = 0.99.

CFI = 0.99. We also created separate delinquency and aggression factors comprised of hierarchical latent factors with loadings on parent and teacher reported aggression and delinquency scales. CFA, as opposed to EFA, were conducted for aggression and delinquency because these are well-established characteristics of antisocial behavior [70, 71]. Model fit for the aggression, $\chi^2(729) = 806.45$, $p = .02$, RMSEA = 0.03, CFI = 0.99, and delinquency, $\chi^2(759) = 855.10$, $p = .01$, RMSEA = 0.03, CFI = 0.98, factors were good.

## Statistical analyses

Statistical analyses were conducted in Mplus version 8.4 [66] utilizing structural equation modeling. As mentioned above, a non-significant $\chi^2$ statistic, RMSEA < .06, and CFI > .95 were indicators of good model fit. The weighted least square mean and variance (WLSMV) estimation method was used, since all analyses included ordinal variables, and missing data were handled using pairwise deletion. Data from individuals within the same family were analyzed using the TYPE = COMPLEX option, which accounts for nonindependence when calculating standard errors and model fit. To correct for multiple testing, we used the False Discovery Rate (FDR) for all regression analyses, which controls for the expected proportion of falsely rejected null hypotheses [72, 73]. For all models, we used an alpha level of .05.

For Aim 1, we evaluated associations between SES, ND, and general ASB (measured by a hierarchical latent variable of parent and teacher reported ASB) with structural equation modeling. For models with nominally significant associations (i.e., $p < .05$ prior to correcting for multiple testing), we tested whether associations are best explained by environmental mediation versus passive $r$GE. Passive $r$GE would be indicated if associations are significantly higher in nonadoptive, compared to adoptive families. Environmental mediation would be suggested by any significant associations in adoptive families. We also examined differences in associations across sex assigned at birth. To examine differences across sex and adoption status, alternative models fixing associations to be equal across adoptive status (e.g., adopted and nonadopted girls; adopted and nonadopted boys) or sex (e.g., nonadopted girls and boys; adopted girls and boys) were compared to models where associations were freed across groups, via a $\chi^2$ difference test.

We used latent basis growth modeling (an efficient method to estimate change over time) to examine associations between predictors and ASB initial levels and change (Fig 2) [74]. ASB loadings were fixed to zero and 1.0 for the first and last timepoint, respectively [75]. Intermediate factor loadings represented percent change between first and last timepoint and were freely estimated. The latent intercept factor measured common variance across timepoints and had an unstandardized loading at 1.0 (see S5 Table for growth model parameters). Lastly, we examined associations separated by aggression and delinquency. Models were run separately for aggression and delinquency and effect sizes and significance of associations were compared.

## Results

### Sample description

**Socioeconomic status.** Given that biological parents of adoptees were younger on average (mothers' mean age = 19.4; fathers' mean age = 21.1) than biological parents of nonadoptees or adoptive parents, they had less opportunity for career or educational attainment. Therefore, for the biological parents of adoptees, the adoptees' maternal grandmothers' and grandfathers' (i.e., the adoptees' grandparents) educational attainment and maternal grandfathers' NORC scores were examined. Results of a one-way ANOVA indicated that there were statistically significant differences in composite SES variables across adoptive parents, biological grandparents, and biological parents of nonadoptees (F(2,727) = [50.47], $p < .001$). Tukey's HSD Test

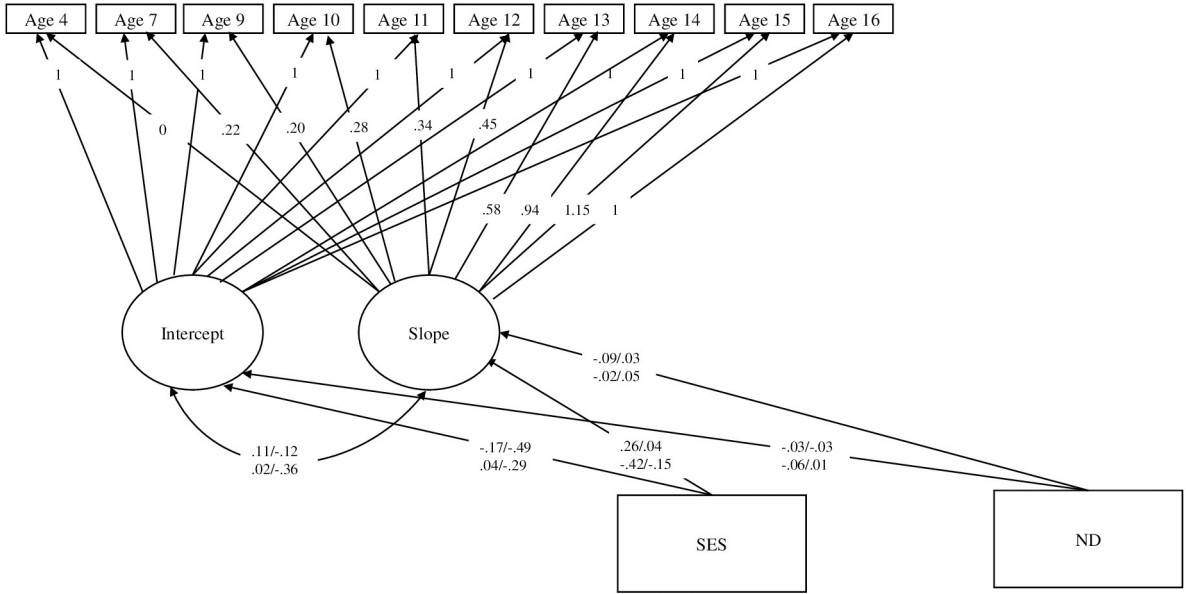

**Fig 2. Parent reported ASB latent growth factor regressed on adoptive parent SES (for adoptees) and Biological Parent SES (for nonadoptees) and ND.** Note: * = freely estimated loadings; "SES" = socioeconomic status; "ND" = neighborhood disadvantage. Unstandardized factor loadings, correlations, and regression coefficients reported. All factor loadings significant at $p < .05$. Model fit: $\chi^2(296) = 368.34$, p = 0.00, RMSEA = .03, CFI = .99, TLI = .99. Estimates presented as: adopted girls/adopted boys nonadopted girls/ nonadopted boys.

for multiple comparisons indicated that biological grandparent SES was significantly lower compared to SES for adoptive parents and biological parents of nonadoptees ($p < .001$), and that there were no differences across the SES of adoptive parents and biological parents of non-adoptees. Cohort effects in education may be partly responsible for the significantly lower SES scores for biological grandparents of adoptees, given that biological grandparents of adoptees were older than adoptive parents and biological parents of nonadoptees.

Education and NORC scores were transformed into *z*-scores, then averaged to create a composite SES variable. Table 1 shows the means and standard deviations for the SES measures and S3 and S4 Tables show correlations between individual measures of SES. Correlations between SES variables were significant at $p < .001$ and ranged from .32 to .60, except for a nonsignificant correlation between mother's education and father's NORC score in nonadopted boys (S4 Table).

**Neighborhood disadvantage.** Overall, the mean ND score, which was the mean of decile scores across the seven indicators, was 4.51 (SD = 1.98). Table 1 shows mean and standard deviations for ND in adoptive versus nonadoptive parents and S3 and S4 Tables show correlations between ND and SES domains. Results of an independent samples between-subjects two-tailed t-test indicated that ND for nonadoptive parents was significantly higher than ND for adoptive parents, $t(297) = [−4.35]$, $p < .001$.

## Correlations

We first examined correlations between SES measures and ND. As expected, SES measures were negatively correlated with ND, although correlations were not significant (S3 and S4 Tables). We then examined correlations between predictors and ASB domains (Table 2). ND and ASB were not significantly correlated, and its association was not consistent in direction

**Table 1. Descriptive Statistics for Parental SES Variables and ND.**

| Adoptees | N | M | SD | Range |
|---|---|---|---|---|
| Biological GF's Number of Years of Schooling Completed | 259 | 13.53 | 2.99 | 5–21 |
| Biological GM's Number of Years of Schooling Completed | 267 | 13.01 | 2.26 | 8–21 |
| Biological GF's NORC Score | 254 | 47.25 | 14.41 | 17.3–81.2 |
| Adoptive Father's Number of Years of Schooling Completed | 233 | 15.67 | 2.46 | 9–21 |
| Adoptive Mother's Number of Years of Schooling Completed | 238 | 14.69 | 2.10 | 10–21 |
| Adoptive Father's NORC Score | 243 | 51.88 | 13.23 | 16.4–81.2 |
| Adoptive Parent's ND | 139 | 4.03 | 2.05 | 0–8.86 |
| Nonadoptees | N | M | SD | Range |
| Biological Father's Number of Years of Schooling Completed | 301 | 15.69 | 2.33 | 6–21 |
| Biological Mother's Number of Years of Schooling Completed | 306 | 14.85 | 2.09 | 8–21 |
| Biological Father's NORC Score | 311 | 51.13 | 12.05 | 19.3–81.2 |
| Biological Parent's ND | 200 | 4.71 | 1.87 | 0–8.9 |

Note: "Biological GF" = biological maternal grandfather of adoptees; "biological GM" = biological maternal grandmother of adoptees; "ND" = neighborhood disadvantage. Limitations in the 1980 census tract data led to decreased Ns of ND compared to SES data.

or magnitude across groups. In nonadoptees, parental SES was significantly negatively correlated with ASB. In general, parental SES was not significantly correlated with ASB in adoptees. Only biological parent SES and aggression were significantly, positively correlated in adopted girls.

Overall, the positive correlations between biological parent SES and ASB domains were opposite in direction from expected in adopted girls. To ensure that correlations were not due to coding errors, we examined correlations between individual ASB measures and individual biological parent SES domains (i.e., maternal grandmother's education, maternal grandfather's education, and maternal grandfather's occupation score). Associations across SES domains and ASB items were positive, consistent with correlations between the biological parent SES composite variable and ASB factors (results available upon request).

## Aim 1: Are SES and ND independently associated with ASB?

In adoptees, we found no evidence for independent effects of predictors on ASB (Table 3). In nonadoptees, biological parent SES nominally predicted ASB in boys, but associations were not significant after FDR correction. Associations between SES and ASB were not significantly different across sex or adoption status, $\Delta\chi^2(2) = 0.488$, $p = .784$ and $\Delta\chi^2(2) = 2.560$, $p = .278$, respectively.

## Aim 2: Do early SES and ND predict change in ASB over time?

Teacher-reported ASB did not show consistent patterns of change over time, likely because ASB was assessed yearly by a different teacher. As such, we only investigated change over time for parent reported items. Parent-reported ASB decreased significantly over time in all groups (S5 Table).

Predictors were not correlated with the intercept (which captures stability with the initial level) of parent reported ASB of parent reported ASB (S6 and S7 Tables) and remained nonsignificant in multiple regression analyses (Table 4). Only SES of the biological parents of nonadoptees was significantly correlated with the ASB slope (in nonadopted girls), such that

**Table 2. Correlations between ND, parent SES and ASB domains.**

| Adoptees | | | | | | | | | | | |
|---|---|---|---|---|---|---|---|---|---|---|---|
| | Biological Parent SES | | | | Adoptive Parent SES | | | | ND | | | |
| | Girls | | Boys | | Girls | | Boys | | Girls | | Boys | |
| | r [CI] | p | r [CI] | p | r [CI] | p | r [CI] | p | r [CI] | p | r [CI] | p |
| ASB Hierarchical Factor | .21 [-.02, .44] | .07 | .01 [-.22, .24] | .93 | .03 [-.19, .25] | .79 | -.07 [-.27, .13] | .49 | -.05 [-.32, .23] | .72 | .15 [-.10, .40] | .23 |
| Aggression Hierarchical Factor | .28* [.05, .52] | .02 | .01 [-.24, .26] | .94 | .07 [-.17, .31] | .55 | -.06 [-.26, .15] | .59 | -.06 [-.35, .24] | .76 | .14 [-.14, .42] | .32 |
| Delinquency Hierarchical Factor | .27 [-.01, .55] | .06 | .04 [-.18, .26] | .70 | -.08 [-.34, .19] | .58 | -.11 [-.31, .10] | .31 | -.06 [-.36, .25] | .72 | .15 [-.08, .38] | .19 |

| Nonadoptees | | | | | | | | | | | |
|---|---|---|---|---|---|---|---|---|---|---|---|
| | Biological Parent SES | | | | ND | | | | | | | |
| | Girls | | Boys | | Girls | | Boys | | | | | |
| | r [CI] | p | r [CI] | p | r [CI] | p | r [CI] | p | | | | |
| ASB Hierarchical Factor | -.18 [-.38, .02] | .55 | -.19* [-.36, -.01] | .04 | .07 [-.16, .31] | .07 | .05 [-.18, .29] | .66 | | | | |
| Aggression Hierarchical Factor | -.17 [-.39, .05] | .12 | -.24* [-.45, -.02] | .03 | .05 [-.21, .30] | .70 | .06 [-.21, .32] | .68 | | | | |
| Delinquency Hierarchical Factor | -.21* [-.41, -.01] | .04 | -.15 [-.34, .04] | .11 | -.01 [-.24, .22] | .93 | .03 [-.18, .25] | .76 | | | | |

*$p < .05$

higher SES was correlated with less decrease in ASB (S6 and S7 Tables). This association remained nominally significant after controlling for ND but was no longer significant after correcting for multiple testing (Table 4). We found no significant sex or adoptive status differences in associations between parental SES and ASB slope, $\Delta\chi^2(2) = 0.947$, p = 0.623 and $\Delta\chi^2(2) = 3.392$, p = 0.066, respectively.

**Table 3. ASB hierarchical factor regressed on parent SES and ND.**

| Adoptees | | | | | | |
|---|---|---|---|---|---|---|
| N = 419 | Adoptive Parent SES | | | ND | | |
| | β [CI] | SE | p | β [CI] | SE | p |
| Girls | .04 [-.18, .25] | .11 | .75 | -.05 [-.33, .23] | .14 | .71 |
| Boys | -.08 [-.28, .13] | .10 | .47 | .16 [-.09, .40] | .13 | .22 |
| N = 389 | Biological Parent SES | | | ND | | |
| | β [CI] | SE | p | β [CI] | SE | p |
| Girls | .21 [-.02, .44] | .12 | .07 | -.04 [-.31, .24] | .14 | .80 |
| Boys | .04 [-.20, .28] | .12 | .75 | .16 [-.10, .42] | .13 | .23 |
| Nonadoptees | | | | | | |
| N = 591 | Biological Parent SES | | | ND | | |
| | β [CI] | SE | p | β [CI] | SE | p |
| Girls | -.17 [-.38, .03] | .10 | .10 | .05 [-.20, .29] | .12 | .71 |
| Boys | -.19* [-.36, -.01] | .09 | .04 | .05 [-.19, .28] | .12 | .70 |

*non-FDR corrected $p < .05$

*Note*: β = standardized regression coefficient; "CI" = confidence interval; "SE" = standard error

Model fit for model examining adoptive parent SES of adoptees and nonadoptive parent SES of nonadoptees: $\chi^2(950)$ = 1046.72, $p = 0.02$, RMSEA = .02, CFI = .99.

Model fit for model examining biological parent SES of adoptees: $\chi^2(440)$ = 479.43, $p = 0.10$, RMSEA = .02, CFI = .99.

**Table 4. Parent-reported ASB slope and intercept regressed on parent SES and ND.**

| | **Adoptees** | | | | | | | | | | | |
|---|---|---|---|---|---|---|---|---|---|---|---|---|
| | Intercept Regression | | | | | | Slope Regression | | | | | |
| *N* = 415 | Adoptive Parent SES | | | ND | | | Biological Parent SES | | | ND | | |
| | β [CI] | SE | *p* | β [CI] | SE | *p* | β [CI] | | | β [CI] | | |
| Girls | -.05 [-.26, .15] | .11 | .62 | .03 [-.21, .26] | .12 | .82 | .01 [-.21, .22] | .11 | .96 | -.17 [-.42, .09] | .13 | .20 |
| Boys | -.19 [-.39, .01] | .10 | .07 | .03 [-.19, .26] | .12 | .78 | .09 [-.19, .37] | .14 | .53 | .21 [-.18, .61] | .20 | .29 |
| *N* = 385 | Biological Parent SES | | | ND | | | Biological Parent SES | | | ND | | |
| | β [CI] | SE | *p* | β [CI] | SE | *p* | β [CI] | SE | *p* | β [CI] | SE | *p* |
| Girls | .13 [-.12, .37] | .12 | .30 | .02 [-.22, .26] | .12 | .87 | .19 [-.07, .45] | .13 | .16 | -.19 [-.47, .08] | .14 | .16 |
| Boys | .04 [-.16, .24] | .10 | .71 | .05 [-.18, .28] | .12 | .68 | -.05 [-.36, .25] | .16 | .73 | .23 [-.21, .66] | .22 | .31 |
| | **Nonadoptees** | | | | | | | | | | | |
| | Intercept Regression | | | | | | Slope Regression | | | | | |
| *N* = 591 | Biological Parent SES | | | ND | | | Biological Parent SES | | | ND | | |
| | β [CI] | SE | *p* | β [CI] | SE | *p* | β [CI] | SE | p | β [CI] | SE | p |
| Girls | .01 [-.21, .24] | .11 | .93 | .15 [-.10, .39] | .12 | .24 | -.28* [-.52, -.05] | .12 | .02 | -.12 [-.42, .18] | .15 | .43 |
| Boys | -.02 [-.22, .17] | .10 | .81 | .04 [-.16, .23] | .10 | .73 | -.14 [-.38, .10] | .12 | .26 | .05 [-.19, .30] | .13 | .67 |

*non-FDR corrected $p < .05$

*Note*: β = standardized regression coefficient; "CI" = confidence interval; "SE" = standard error

Model fit for model examining adoptive parent SES of adoptees and nonadoptive parent SES of nonadoptees: $\chi^2(296) = 368.34$, $p = 0.00$; RMSEA = .03, CFI = .99, TLI = .99.

Model fit for model examining biological parent SES of adoptees: $\chi^2(143) = 170.76$, $p = 0.06$, RMSEA = .03, CFI = 1.00.

## Aim 3: Are there differences in patterns of associations between SES, ND, and aggression and delinquency scales?

**Aggression.** ND was neither significantly correlated nor independently associated with aggression (Table 2). In adopted girls, biological parent SES was positively correlated with aggression. Biological parent SES of adoptees remained an independent predictor after controlling for ND (Table 5). There were no significant sex differences in these associations, $\Delta\chi^2(1) = 1.951$, p = .163. In nonadopted boys, biological parent SES and aggression were significantly, negatively correlated. This association remained nominally significant after controlling for ND. These associations did not significantly differ across sex or adoption status, $\Delta\chi^2(2) = 0.818$, p = .665 and $\Delta\chi^2(2) = 3.301$, p = .192, respectively. No nominally significant associations were significant after correcting for multiple testing.

**Delinquency.** ND was neither significantly correlated nor independently associated with delinquency (Table 2). Only biological parent SES in nonadopted girls was significantly correlated with delinquency. This association remained significant after controlling for ND but was no longer significant after multiple testing correction (Table 5). We found no significant differences across sex or adoption status, $\Delta\chi^2(2)$ 0.357, p = .836 and sex, $\Delta\chi^2(2) = 0.810$, p = .667, respectively.

## Sensitivity and attrition analyses

Participants provided consent for analyses of early ND data many years after initial assessment. We conducted attrition and sensitivity analyses to determine whether missingness for ND predicted differences in early SES, ASB, or associations across predictors. Independent samples t-tests demonstrated that individuals with ND data had significantly higher parental SES $t(488) = [2.83]$, $p = .005$, but no significant differences in ASB, $t(735) = [.23]$, $p = .82$. Additionally,

**Table 5. Aggression and delinquency factors regressed on parent SES and ND.**

| | | | | | | |
|---|---|---|---|---|---|---|
| **Adoptees** | | | | | | |
| Aggression Factor | | | | | | |
| N = 389 | Adoptive Parent SES | | | ND | | |
| | β [CI] | SE | p | β [CI] | SE | p |
| Girls | .08 [-.16, .32] | .12 | .52 | -.06 [-.37, .24] | .16 | .68 |
| Boys | -.06 [-.27, .15] | .11 | .57 | .14 [-.13, .41] | .14 | .31 |
| N = 356 | Biological Parent SES | | | ND | | |
| | β [CI] | SE | p | β [CI] | SE | p |
| Girls | .28* [.05, .51] | .12 | .02 | -.05 [-.34, .25] | .15 | .75 |
| Boys | .04 [-.22, .29] | .13 | .79 | .15 [-.14, .44] | .15 | .31 |
| Delinquency Factor | | | | | | |
| N = 389 | Adoptive Parent SES | | | ND | | |
| | β [CI] | SE | p | β [CI] | SE | p |
| Girls | -.07 [-.34, .19] | .14 | .60 | -.05 [-.35, .25] | .15 | .75 |
| Boys | -.11 [-.32, .10] | .11 | .30 | .16 [-.07, .39] | .12 | .18 |
| N = 356 | Biological Parent SES | ND | | | | |
| | β [CI] | SE | p | β [CI] | SE | p |
| Girls | .27 [-.01, .55] | .14 | .06 | -.04 [-.33, .26] | .15 | .81 |
| Boys | .08 [-.15, .31] | .12 | .50 | .17 [-.07, .41] | .12 | .16 |
| **Nonadoptees** | | | | | | |
| Aggression Factor | | | | | | |
| N = 591 | Biological Parent SES | | | ND | | |
| | β [CI] | SE | p | β [CI] | SE | p |
| Girls | -.17 [-.39, .05] | .12 | .14 | .03 [-.24, .29] | .13 | .85 |
| Boys | -.23* [-.45, -.02] | .11 | .03 | .05 [-.22, .31] | .13 | .73 |
| Delinquency Factor | | | | | | |
| N = 591 | Biological Parent SES | | | ND | | |
| | β [CI] | SE | p | β [CI] | SE | p |
| Girls | -.22* [-.42, -.02] | .10 | .03 | -.04 [-.28, .20] | .12 | .73 |
| Boys | -.15 [-.33, .04] | .09 | .11 | .03 [-.19, .25] | .11 | .80 |

*non-FDR corrected $p < .05$

*Note:* β = standardized regression coefficient; "CI" = confidence interval; "SE" = standard error

Aggression: model fit for model examining adoptive parent SES of adoptees and nonadoptive parent SES of nonadoptees: $\chi^2(873) = 961.81$, $p = 0.02$, RMSEA = 0.02, CFI = 0.99.

Aggression: model fit for model examining biological parent SES: $\chi^2(415) = 440.74$, $p = 0.18$, RMSEA = 0.02, CFI = 1.00.

Delinquency: model fit for model examining adoptive parent SES of adoptees and nonadoptive parent SES of nonadoptees: $\chi^2(903) = 987.74$, $p = 0.03$, RMSEA = 0.02, CFI = 0.98.

Delinquency: model fit for model examining biological parent SES: $\chi^2(425) = 434.85$, $p = 0.36$, RMSEA = 0.01, CFI = 1.00.

we conducted analyses including only individuals with ND data. Results were similar in analyses including only individuals with ND data and those including all participants (S8–S14 Tables), including slope estimates for latent growth curve models and effect sizes for regression of ASB slope and hierarchical ASB factor on SES and ND.

## Discussion

The present study utilized a longitudinal adoption design to examine the etiology of associations between SES, ND, and ASB. We aimed to replicate previous findings suggesting

independent influences of SES and ND on ASB and to add to extant literature by addressing whether these are due to environmental mediation, passive $r$GE, or a combination of the two processes. We examined the influence of predictors on ASB over time, tested for sex differences in all associations, and evaluated the magnitude of associations between SES, ND, and aggressive versus delinquent ASB.

Contrary to prior research [16, 21, 23, 24, 27, 29–31], our analyses for Aim 1 did not demonstrate that ND independently predicted ASB in any group. Effect sizes were small, not significant, and inconsistent in direction. Additionally, most previous research tested associations between ND assessed in middle childhood and ASB in middle childhood to adolescence, although one study did find long-lasting effects of ND assessed during infancy on later ASB [32]. The nonsignificant associations between ND and ASB may be in part explained by the limited variability of early ND in the present sample, as the association between ND and ASB may be strongest at highest levels of ND [30].

In contrast to our results regarding ND, associations between SES and ASB in nonadoptees were consistent with direction of results in the extant literature [12, 13, 23, 24]. Overall, SES for biological parents of nonadoptees was negatively associated with ASB. Additionally, effect sizes in nonadoptees (standardized betas ranging from -.15 to -.28) were comparable to or larger than those reported in existing studies, **a**s a meta-analyses of associations between SES and ASB have found small but significant overall effects [12, 13]. In general, children whose parents had lower education and occupation scores were at higher risk for overall ASB in addition to aggressive and delinquent behavior. These results further illustrate the negative contribution of lower SES on the psychological health of children and adolescents and demonstrate that this influence can begin as early as infancy. Additionally, prior research has demonstrated that familial factors generally have a greater magnitude of influence on ASB compared to neighborhood level factors [24] and our results may reflect this phenomenon.

Alignment between results of the present study and prior research did not extend to results in adoptees. The direction of effects between adoptive parent SES and ASB were inconsistent, and associations were not significant. Counterintuitively, associations between SES of biological parents of adoptees and ASB were generally positive in direction and nominally significant (specifically for associations between biological parent SES and ASB in adopted girls). These results should be regarded with caution and interpreted in the context of the number of tests conducted. If these results indicate a broader pattern of positive associations between SES and ASB between biological parents and their children, we would likely have seen similar associations between nonadoptive parents and nonadoptees.

Regarding Aims 1 and 4, we found no evidence for differences in associations across sex or adoptees (in which adopted children and adoptive parents share only their environment) and nonadoptees (in which nonadopted children share both genes and environment with their parents). Overall, our findings regarding sex differences align with our hypotheses and prior research [11, 41, 76, 77]. However, our results did not replicate a previous adoption study that observed sex differences in the mechanisms by which SES influenced ASB (i.e., stronger environmental influences for boys than girls; [78] and we were unable to conclusively determine whether associations were primarily due to environmental influences or passive $r$GE.

ASB significantly decreased over time in our sample, consistent with expectations [79, 80], but we did not find evidence that differences in decreases in ASB were due to SES or ND (Aim 3). Higher SES for biological parents of nonadoptees nominally predicted less ASB decline in girls only. Prior research on this topic is minimal and found that high ND and lower SES predicted less decline and increase in ASB, respectively [21, 57]. Additional research on predictors of change in ASB is needed. In addition, we found similar effect sizes across aggressive versus nonaggressive behaviors (Aim 3).

Importantly, all results should be interpreted with caution; although several of our results were nominally significant, associations did not remain significant after controlling for multiple testing. As such, we also would not necessarily expect significant differences in associations across sex or adoptive status. Thus, we cannot make definitive conclusions about the genetic versus environmental etiology of associations between predictors and ASB (Aim 1). We also cannot conclude confidently whether influence of SES or ND on ASB differs across sex.

## Limitations

There are several limitations to consider when interpreting these results. First, it is often assumed correctly that adoption studies have a restricted range of environmental influences provided by adoptive parents. Potential adoptive parents are often precluded from adopting if they do not meet thresholds of financial security, mental health, and marital stability. Of those environmental factors most restricted in adoption studies, SES is among the top, with one study estimating an 18% reduction in variance [81]. Interestingly, the same study showed that ASB in adoptive parents was much less restricted than SES, with only a 7–8% reduction in variance for adoptive parents. Second, only paternal occupational status was examined since the adoption agencies required one parent be an at-home parent and few mothers worked outside of the home when the study was initiated, but SES estimates may be low for families with mothers working outside of the home. Third, measures used for SES and ND were obtained in general when subjects were less than a year old. Changes in familial SES or ND may have occurred over the course of the study and our estimates do not capture how those changes or SES and ND in later childhood or adolescence influence ASB, although changes in SES and ND may play a role in ASB change over time. Children may be more likely to engage in delinquent behaviors during years when their parents' SES is lower [52] and ASB may decrease if children are they are relocated from high to low ND environments [52, 53]. Research addressing this issue by assessing familial SES and ND along with ASB prospectively and regularly is needed.

In addition, other measures of ND may better capture factors influencing child ASB, as prior research has demonstrated that the magnitude of the associations between ND and child health [82] and externalizing outcomes [83] depend on the ND metric examined and covariates included, such as family level SES. Lastly, in adoptive families, the influence of adoptive familial SES [58], but likely not ND [84], on child ASB may be moderated by biological parent ASB. However, we were unable to test for gene-environment interaction in the present study, due to lack of a standard ASB assessment in biological parents of adoptees.

## Strengths

The CAP is an ideal adoption study to examine putative environmental effects because it addresses potential limitations of adoption designs [85]. In the CAP, selective placement (e.g., resemblance between adoptive and biological parents that inflates estimates of genetic and environmental influences; [86, 87] is negligible for measures of education and SES [60]. Adoptees in the study were placed in adoptive homes within 29 days of birth, mitigating confounded genetic and environmental influences.

## Conclusions

Despite limitations, the present study provides a unique contribution to the literature on the mechanisms involved in associations between SES, ND, and ASB. We found nominally significant negative associations between ASB and SES in nonadoptive families, although associations were no longer significant after correcting for multiple testing. Associations between ND

and ASB were nonsignificant and inconsistent in direction. In this study, ND at an early age was not a reliable predictor of ASB. We found no evidence for differences in associations across sex, and the effects sizes of the associations between SES/ND and the aggression and delinquency scales were similar. Although our study's findings were inconclusive, in nonadoptees, results align in effect size and direction with prior research indicating that individuals with lower SES are at higher risk for ASB. Results provide support for the potential impact of familial poverty on child psychological health and as income inequality rises globally [88], supporting families via effective social programs may mitigate poverty's impact on negative child outcomes [21, 89].

## Supporting information

**S1 Table. Frequencies for parent and teacher reported externalizing scales.**
(DOCX)

**S2 Table. Descriptive statistics for parent and teacher reported aggression and delinquency scales.**
(DOCX)

**S3 Table. Correlations between SES variables and ND in adoptees.**
(DOCX)

**S4 Table. Correlations between SES variables and ND in nonadoptees.**
(DOCX)

**S5 Table. Parent reported ASB slope, intercept, and correlations between slope and intercept.**
(DOCX)

**S6 Table. Correlations between parent reported ASB intercept and slope, ND, and biological and adoptive parent SES in adoptees.**
(DOCX)

**S7 Table. Correlations between parent reported ASB intercept, slope, and biological Parent SES in nonadoptees.**
(DOCX)

**S8 Table. ASB hierarchical factor regressed on adoptive and biological parent SES and ND in adoptees: Individuals with ND data only.**
(DOCX)

**S9 Table. ASB hierarchical factor regressed on biological parent SES and ND in nonadoptees: Individuals with ND data only ($N$ = 365).**
(DOCX)

**S10 Table. Parent reported ASB slope, intercept, and correlations between slope and intercept: Individuals with ND data only ($N$ = 428).**
(DOCX)

**S11 Table. Parent reported ASB intercept regressed on biological and adoptive parent SES and ND in adoptees: Individuals with ND data only.**
(DOCX)

**S12 Table. Parent reported ASB slope regressed on biological and adoptive parent SES and ND in adoptees: Individuals with ND data only.**
(DOCX)

**S13 Table. Parent reported ASB intercept regressed on biological parent SES and ND in nonadoptees: Individuals with ND data only ($N$ = 365).**
(DOCX)

**S14 Table. Parent reported ASB slope regressed on biological parent SES and ND in nonadoptees: Individuals with ND data only ($N$ = 365).**
(DOCX)

## Acknowledgments

The authors would like to acknowledge the individuals who participated in this study.

## Author Contributions

**Conceptualization:** Shelley A. Gresko, Laura K. Hink, Soo Hyun Rhee.

**Data curation:** Shelley A. Gresko, Laura K. Hink, Robin P. Corley, Chandra A. Reynolds, Elizabeth Muñoz.

**Formal analysis:** Shelley A. Gresko, Laura K. Hink.

**Methodology:** Shelley A. Gresko, Soo Hyun Rhee.

**Project administration:** Robin P. Corley.

**Supervision:** Soo Hyun Rhee.

**Validation:** Shelley A. Gresko, Laura K. Hink, Chandra A. Reynolds, Elizabeth Muñoz.

**Visualization:** Shelley A. Gresko, Laura K. Hink.

**Writing – original draft:** Shelley A. Gresko, Laura K. Hink, Soo Hyun Rhee.

**Writing – review & editing:** Shelley A. Gresko, Robin P. Corley, Chandra A. Reynolds, Elizabeth Muñoz, Soo Hyun Rhee.

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
