## [Decision Letter · Decision Letter 0]

26 Jan 2024

PONE-D-23-32462An examination of early socioeconomic status and neighborhood disadvantage as independent predictors of antisocial behavior: a longitudinal adoption studyPLOS ONE

Dear Dr. Gresko,

Thank you for submitting your manuscript to PLOS ONE. After careful consideration, we feel that it has merit but does not fully meet PLOS ONE’s publication criteria as it currently stands. Therefore, we invite you to submit a revised version of the manuscript that addresses the points raised during the review process.

I have received the report of the reviewers. You can find below their coments. The manuscript needs a MAYOR REVISION.

I recommend that the authors carefully address the major and minor comments outlined above and provide a revised version of the manuscript. It would be helpful if the authors include a detailed response letter indicating how each comment has been addressed in the revised manuscript. Furthermore, to make sure that the manuscript adheres to the journal's guidelines and formatting requirements, please, check the “Instructions for the authors” document: https://journals.plos.org/plosone/s/submission-guidelines#loc-financial-disclosure-statement

Please submit your revised manuscript by Mar 11 2024 11:59PM.  If you will need more time than this to complete your revisions, please reply to this message or contact the journal office at plosone@plos.org. Please include the following items when submitting your revised manuscript:A rebuttal letter that responds to each point raised by the academic editor and reviewer(s). You should upload this letter as a separate file labeled 'Response to Reviewers'.A marked-up copy of your manuscript that highlights changes made to the original version. You should upload this as a separate file labeled 'Revised Manuscript with Track Changes'.An unmarked version of your revised paper without tracked changes. You should upload this as a separate file labeled 'Manuscript'.

We look forward to receiving your revised manuscript.

Kind regards,

Bárbara Oliván-Blázquez, Ph.D.

Academic Editor

PLOS ONE

 Journal requirements: 1. When submitting your revision, we need you to address these additional requirements. Please ensure that your manuscript meets PLOS ONE's style requirements, including those for file naming. The PLOS ONE style templates can be found at https://journals.plos.org/plosone/s/file?id=wjVg/PLOSOne_formatting_sample_main_body.pdf and https://journals.plos.org/plosone/s/file?id=ba62/PLOSOne_formatting_sample_title_authors_affiliations.pdf. 2. Note from Emily Chenette, Editor in Chief of PLOS ONE, and Iain Hrynaszkiewicz, Director of Open Research Solutions at PLOS: Did you know that depositing data in a repository is associated with up to a 25% citation advantage (https://doi.org/10.1371/journal.pone.0230416)? If you’ve not already done so, consider depositing your raw data in a repository to ensure your work is read, appreciated and cited by the largest possible audience. You’ll also earn an Accessible Data icon on your published paper if you deposit your data in any participating repository (https://plos.org/open-science/open-data/#accessible-data). 3. We note that you have indicated that there are restrictions to data sharing for this study. PLOS only allows data to be available upon request if there are legal or ethical restrictions on sharing data publicly. For more information on unacceptable data access restrictions, please see http://journals.plos.org/plosone/s/data-availability#loc-unacceptable-data-access-restrictions.  Before we proceed with your manuscript, please address the following prompts: a) If there are ethical or legal restrictions on sharing a de-identified data set, please explain them in detail (e.g., data contain potentially identifying or sensitive patient information, data are owned by a third-party organization, etc.) and who has imposed them (e.g., a Research Ethics Committee or Institutional Review Board, etc.). Please also provide contact information for a data access committee, ethics committee, or other institutional body to which data requests may be sent. b) If there are no restrictions, please upload the minimal anonymized data set necessary to replicate your study findings to a stable, public repository and provide us with the relevant URLs, DOIs, or accession numbers. For a list of recommended repositories, please seehttps://journals.plos.org/plosone/s/recommended-repositories. You also have the option of uploading the data as Supporting Information files, but we would recommend depositing data directly to a data repository if possible. We will update your Data Availability statement on your behalf to reflect the information you provide. 4. Please include your full ethics statement in the ‘Methods’ section of your manuscript file. In your statement, please include the full name of the IRB or ethics committee who approved or waived your study, as well as whether or not you obtained informed written or verbal consent. If consent was waived for your study, please include this information in your statement as well.  5. Please include a copy of Table S16 in Supporting Information
 which you refer to in your text on page 23.

Reviewers' comments:

Reviewer's Responses to Questions

**Comments to the Author**

1. Is the manuscript technically sound, and do the data support the conclusions?

Reviewer #1: Partly

Reviewer #2: Yes

2. Has the statistical analysis been performed appropriately and rigorously? 

Reviewer #1: No

Reviewer #2: Yes

3. Have the authors made all data underlying the findings in their manuscript fully available?

Reviewer #1: Yes

Reviewer #2: No

4. Is the manuscript presented in an intelligible fashion and written in standard English?

Reviewer #1: Yes

Reviewer #2: Yes

5. Review Comments to the Author

Reviewer #1: This study shares important information about the potential impact of socioeconomic status on children's psychological health.

An extensive review of the topic has been carried out as a background, however, reviewing the bibliography used, I consider that it would be necessary to incorporate more current existing bibliography. It is true that it is a project started in 1975, but if the purpose is to give it scope and provide us with evidence of the present, it would be necessary to incorporate more recent information.

On the other hand, in Line 46, if "" is part of the objective of the study, I do not consider it so relevant to put this information at the beginning of the introduction without having given justifying arguments.

Methodologically, it is a study that uses very complex statistical analyses. However, I have doubts about this methodology:

--There is a lack of a contribution of ethical considerations in methodology, which specifies the ethical committee and how the data have been treated, which in this study precisely involving minors has to be very well argued.

-Line 160 indicates that they come from data from a longitudinal study. started in 1975 and recruited as indicated on line 171 until 1987.

Being a longitudinal study, it is not indicated if or when these people were recruited. Were they monitored during the recruitment period?

Reviewing the articles you cite (Plomin and DeFries (55) and Rhea et al. (49). ), they also do not specify when follow-up evaluations are done.

On the other hand, line 191 specifies that the data has been accessed in 2021.

I consider that it would be necessary to better explain what date the data presented in this manuscript corresponds to.

In measures, a description of the sample has been made in terms of the socioeconomic variables (From line 202 to 221, including table 1), of the Neighborhood disadvantage variables (Line 242-246). I consider this information provided to be not so relevant in methodology but rather as presentation and description of the sample at the beginning of the results section.

As in the introduction, an extensive search has been carried out in the discussion, but also outdated in terms of publication dates.

Reviewer #2: General Assessment:

The introduction effectively addresses various aspects related to Antisocial Behavior (ASB), focusing on potential environmental predictors such as parental socioeconomic status and neighborhood disadvantage. It explores the associations between socioeconomic status (SES), neighborhood disadvantage (ND), and ASB across different genders and types of antisocial behaviors. Additionally, the introduction highlights the use of adoption studies to differentiate between passive genetic influences (passive rGE) and environmental mediation in understanding ASB. The topic is highly relevant to the field, and the research question is well-defined. The study investigated the influence of early socioeconomic status (SES) and neighborhood disadvantage (ND) on antisocial behavior (ASB) using a longitudinal adoption design. The research, based on data from the Colorado Adoption Project, found that lower SES was a significant predictor of ASB in nonadoptive families, while ND did not show a significant association with ASB. The study did not provide conclusive evidence regarding the genetic versus environmental factors influencing the relationship between SES, ND, and ASB, as associations were not consistently significant across sex or adoptive status. However, in nonadopted individuals, lower SES was consistently associated with an increased risk of ASB, aligning with previous research findings.

First of all, to make sure that the manuscript adheres to the journal's guidelines and formatting requirements, please, check the “Instructions for the authors” document:

https://journals.plos.org/plosone/s/submission-guidelines#loc-financial-disclosure-statement

Apart from that, please consider the below comments to strengthen the study as reported, which reflect my responses to general checklists:

Major Comments:

• Line 182. In terms of racial diversity, could you provide more information on the distribution within the five reported categories OR whether there were any notable differences in outcomes based on race?

• Line 200. The paragraph mentions that only paternal occupational status was examined due to limited mothers working outside the home at the study's initiation. Could you provide more context on this decision and its potential impact on the comprehensiveness of the SES assessment?

• Line 259. The decision to conduct confirmatory factor analyses (CFA) for examining aggression, delinquency, and overall Antisocial Behavior (ASB) is mentioned. Could you provide insights into the specific reasons or theoretical justifications for choosing CFA over exploratory factor analysis (EFA) in this study?

• Line 476. The mention of potential changes in SES and ND in later childhood or adolescence raises questions about the dynamics of these factors over time. Could you discuss how these potential changes might influence the study's results and interpretations, and suggest ways to address this limitation in future research?

Minor Comments:

• In line 260, page 13. There is a symbol that is not displayed well. Same in line 280, 299.

Overall, the manuscript has the potential to make a valuable contribution to the scientific community. With appropriate revisions, it could be considered suitable for publication in Plos One.

Recommendation:

I recommend that the authors carefully address the major and minor comments outlined above and provide a revised version of the manuscript. It would be helpful if the authors include a detailed response letter indicating how each comment has been addressed in the revised manuscript.

6. PLOS authors have the option to publish the peer review history of their article (what does this mean?). If published, this will include your full peer review and any attached files.

Reviewer #1: **Yes: **Fárima Méndez-López

Reviewer #2: No

---

## [Author Response · Author response to Decision Letter 0]

29 Feb 2024

We have ensured that our manuscript meets PLOS ONE’s style requirements, including those for file naming.

A portion of our data is currently available through the Harvard Dataverse at https://doi.org/doi:10.7910/DVN/BCDSEU (Plomin et al., 2022) and access can be requested by contacting the Manager of Operations for the Henry A. Murray Research Archive, Institute for Quantitative Social Sciences, 1737 Cambridge St, Cambridge, MA 02138, USA., mra@help.hmdc.harvard.edu. Researchers who meet the criteria for access to confidential data may apply for access to data not yet added to the Harvard Dataverse by contacting the corresponding author (shelley.gresko@colorado.edu) and the University of Colorado Boulder Institutional Review Board (irbadmin@colorado.edu).

If there are ethical or legal restrictions on sharing a de-identified data set, please explain them in detail (e.g., data contain potentially identifying or sensitive patient information, data are owned by a third-party organization, etc.) and who has imposed them (e.g., a Research Ethics Committee or Institutional Review Board, etc.). Please also provide contact information for a data access committee, ethics committee, or other institutional body to which data requests may be sent.

We have provided the following information on ethical restrictions on sharing a de-identified data set: “Given that the data contain potentially identifying and sensitive patient information, data cannot be shared publicly because of restrictions put in place by the University of Colorado Boulder Institutional Review Board. Unidentifiable patient data are in the process of being added to the Harvard Dataverse repository. A portion of this data is currently available through the Harvard Dataverse at https://doi.org/doi:10.7910/DVN/BCDSEU (Plomin et al., 2022) and access can be requested by contacting the Manager of Operations for the Henry A. Murray Research Archive, Institute for Quantitative Social Sciences, 1737 Cambridge St, Cambridge, MA 02138, USA., mra@help.hmdc.harvard.edu. Researchers who meet the criteria for access to confidential data may apply for access to data not yet added to the Harvard Dataverse by contacting the corresponding author (shelley.gresko@colorado.edu) and the University of Colorado Boulder Institutional Review Board (irbadmin@colorado.edu).”

Plomin, R., DeFries, J. C., & Fulker, D. W. (2022). Colorado Adoption Project, 1976-1989 Version V1) [longitudinal, field study, hereditary]. Harvard Dataverse. https://doi.org/doi:10.7910/DVN/BCDSEU

In our Methods section on page 9, lines 187-191, we include the following: “This project is approved by the University of Colorado Boulder Office of Research Integrity’s Institutional Review Board (protocol number 14-0421). Parents completed written consent and children aged seven and older provided written or verbal assent and children provided written consent at age 16. Only participants who provided consent for researchers to geocode first addresses were included in this study.

5. Please include a copy of Table S16 in Supporting Information which you refer to in your text on page 23.

Thank you for drawing attention to this discrepancy. The reference to Table S16 in Supporting Information was an error and has been deleted.

5. Review Comments to the Author

Reviewer #1: This study shares important information about the potential impact of socioeconomic status on children's psychological health.

An extensive review of the topic has been carried out as a background, however, reviewing the bibliography used, I consider that it would be necessary to incorporate more current existing bibliography. It is true that it is a project started in 1975, but if the purpose is to give it scope and provide us with evidence of the present, it would be necessary to incorporate more recent information.

We have added the following studies to include more contemporary references in the introduction section:

13. Peverill M, Dirks MA, Narvaja T, Herts KL, Comer JS, McLaughlin KA. Socioeconomic status and child psychopathology in the United States: A meta-analysis of population-based studies. Clin Psychol Rev. 2021 Feb 1;83:101933. 

14. Chang LY, Wang MY, Tsai PS. Neighborhood disadvantage and physical aggression in children and adolescents: A systematic review and meta-analysis of multilevel studies. Aggress Behav. 2016;42(5):441–54. 

29. Choi JK, Kelley MS, Wang D. Neighborhood Characteristics, Maternal Parenting, and Health and Development of Children from Socioeconomically Disadvantaged Families. Am J Community Psychol. 2018;62(3–4):476–91.

33. Latham RM, Arseneault L, Alexandrescu B, Baldoza S, Carter A, Moffitt TE, et al. Violent experiences and neighbourhoods during adolescence: understanding and mitigating the association with mental health at the transition to adulthood in a longitudinal cohort study. Soc Psychiatry Psychiatr Epidemiol. 2022 Dec 1;57(12):2379–91. 

41. Burt SA. The Genetic, Environmental, and Cultural Forces Influencing Youth Antisocial Behavior Are Tightly Intertwined. Annu Rev Clin Psychol. 2022;18(1):155–78.

On the other hand, in Line 46, if "" is part of the objective of the study, I do not consider it so relevant to put this information at the beginning of the introduction without having given justifying arguments.

We have removed the following sentence “The present study aims to examine whether early childhood socioeconomic status measured in infancy and neighborhood disadvantage independently contribute to level and change of ASB and whether these influences are best explained by gene¬–environment correlation or environmental mediation, using a longitudinal adoption design.” in the first paragraph of the introduction per the reviewer’s suggestion.

Methodologically, it is a study that uses very complex statistical analyses. However, I have doubts about this methodology:

--There is a lack of a contribution of ethical considerations in methodology, which specifies the ethical committee and how the data have been treated, which in this study precisely involving minors has to be very well argued.

On page 9, lines 187-189, we clarify the following related to ethical considerations: “This project is approved by the University of Colorado Boulder Office of Research Integrity’s Institutional Review Board (protocol number 14-0421). Parents completed written consent and children aged seven and older provided written or verbal assent and children provided written consent at age 16.”

-Line 160 indicates that they come from data from a longitudinal study. started in 1975 and recruited as indicated on line 171 until 1987.

Being a longitudinal study, it is not indicated if or when these people were recruited. Were they monitored during the recruitment period?

Reviewing the articles you cite (Plomin and DeFries (55) and Rhea et al. (49). ), they also do not specify when follow-up evaluations are done.

On pages 8-9, lines 169-177, we clarify when parents and siblings were recruited and have added additional information: Prospective recruitment began on January 1st, 1976, and ended on September 30th, 1987. Biological mothers of adoptees were recruited from two large adoption agencies in Colorado. Their children were placed into their adoptive family homes within one year after birth. Social workers matched adoptees with adoptive families on non-proximity of location and similarity of height between adoptive and biological parents. No additional explicit selective placement practices were followed. Seventy-five percent of adoptive parents recruited for the study agreed to participate. Matched control families were recruited from local hospitals. Sibling enrollment and recruitment ended with the last longitudinally followed sibling. 

On page 9, lines 177-180, we specify how often follow up evaluations were completed and have added information about age at assessment: “Biological and adoptive parents were assessed with a comprehensive battery of psychological measures when children were one year old or younger. Adoptees and nonadoptees were assessed approximately annually either through home or lab visits or telephone interviews, from age one year to 16 years. 

On the other hand, line 191 specifies that the data has been accessed in 2021.

I consider that it would be necessary to better explain what date the data presented in this manuscript corresponds to.

On page 9, lines 193-196, we have clarified that the raw data collected prospectively beginning in 1976 and was accessed by the authors of the manuscript in 2021 and 2023: “Data were collected beginning in 1976. The socioeconomic status and antisocial behavior data were accessed by authors on April 27th, 2021, and the neighborhood disadvantage data were finalized and accessed on January 20th, 2023 (see Methods section for more details on the neighborhood disadvantage variable).”

In measures, a description of the sample has been made in terms of the socioeconomic variables (From line 202 to 221, including table 1), of the Neighborhood disadvantage variables (Line 242-246). I consider this information provided to be not so relevant in methodology but rather as presentation and description of the sample at the beginning of the results section.

We have moved the sample description for SES and ND variables into the results section, under the heading “Sample Description” (pages 14-15).

As in the introduction, an extensive search has been carried out in the discussion, but also outdated in terms of publication dates.

We have added the following studies to include more contemporary references in the discussion section:

13. Peverill M, Dirks MA, Narvaja T, Herts KL, Comer JS, McLaughlin KA. Socioeconomic status and child psychopathology in the United States: A meta-analysis of population-based studies. Clin Psychol Rev. 2021 Feb 1;83:101933. 

14. Chang LY, Wang MY, Tsai PS. Neighborhood disadvantage and physical aggression in children and adolescents: A systematic review and meta-analysis of multilevel studies. Aggress Behav. 2016;42(5):441–54. 

41. Burt SA. The Genetic, Environmental, and Cultural Forces Influencing Youth Antisocial Behavior Are Tightly Intertwined. Annu Rev Clin Psychol. 2022;18(1):155–78.

85. Burt SA, Klump KL, Gorman-Smith D, Neiderhiser JM. Neighborhood Disadvantage Alters the Origins of Children’s Nonaggressive Conduct Problems. Clin Psychol Sci. 2016 May 1;4(3):511–26.

Reviewer #2: General Assessment:

The introduction effectively addresses various aspects related to Antisocial Behavior (ASB), focusing on potential environmental predictors such as parental socioeconomic status and neighborhood disadvantage. It explores the associations between socioeconomic status (SES), neighborhood disadvantage (ND), and ASB across different genders and types of antisocial behaviors. Additionally, the introduction highlights the use of adoption studies to differentiate between passive genetic influences (passive rGE) and environmental mediation in understanding ASB. The topic is highly relevant to the field, and the research question is well-defined. The study investigated the influence of early socioeconomic status (SES) and neighborhood disadvantage (ND) on antisocial behavior (ASB) using a longitudinal adoption design. The research, based on data from the Colorado Adoption Project, found that lower SES was a significant predictor of ASB in nonadoptive families, while ND did not show a significant association with ASB. The study did not provide conclusive evidence regarding the genetic versus environmental factors influencing the relationship between SES, ND, and ASB, as associations were not consistently significant across sex or adoptive status. However, in nonadopted individuals, lower SES was consistently associated with an increased risk of ASB, aligning with previous research findings.

First of all, to make sure that the manuscript adheres to the journal's guidelines and formatting requirements, please, check the “Instructions for the authors” document:

https://journals.plos.org/plosone/s/submission-guidelines#loc-financial-disclosure-statement

Apart from that, please consider the below comments to strengthen the study as reported, which reflect my responses to general checklists:

Major Comments:

• Line 182. In terms of racial diversity, could you provide more information on the distribution within the five reported categories OR whether there were any notable differences in outcomes based on race?

We have included more specific demographic information regarding percentage of participants who belonged to each race on page 9, lines 180-184:

“The sample is over 90% White and is demographically representative of the Denver Metropolitan area at the time of recruitment (55). Participants self-reported their race as one of five categories: Alaskan Native/American Indigenous (1.4%), Asian (4.7), Black (0.6%), White (91.6%), and more than one race (0.9%), or unknown or not reported (0.9%).”

Given that the number of non-White subjects is very small, we do not have the power to detect any significant differences in outcomes based on race.

• Line 200. The paragraph mentions that only paternal occupational status was examined due to limited mothers working outside the home at the study's initiation. Could you provide more context on this decision and its potential impact on the comprehensiveness of the SES assessment?

On page 10, lines 205-209, we have provided additional context regarding the decision to include only paternal occupational status: “Only paternal occupational status was examined; given that the adoption agencies required that one parent be an at-home parent and given the limited number of mothers working outside of the home when the study was initiated, including occupational ratings for mothers would not have meaningfully added to the assessment of familial SES during infancy.”

We have also included this in the limitations section on page 30, lines 492-495: “Second, only paternal occupational status was examined since the adoption agencies required one parent be an at-home parent and few mothers worked outside of the home when the study was initiated, but SES estimates may be low for families with mothers working outside of the home.”

• Line 259. The decision to conduct confirmatory factor analyses (

---

## [Decision Letter · Decision Letter 1]

24 Mar 2024

An examination of early socioeconomic status and neighborhood disadvantage as independent predictors of antisocial behavior: a longitudinal adoption study

PONE-D-23-32462R1

Dear Dr. Gresko,

We’re pleased to inform you that your manuscript has been judged scientifically suitable for publication and will be formally accepted for publication once it meets all outstanding technical requirements.

Kind regards,

Bárbara Oliván-Blázquez, Ph.D.

Academic Editor

PLOS ONE

Additional Editor Comments (optional):

Dear Shelley Gresko,

I'm pleased to inform you that your manuscript has been accepted for being published in Pthe journal Plos One.

The authors have the responded to the reviewers' comments, and the manuscript has the clarity, quality, and innovation to be published in Plos One.

congratulations.

Reviewers' comments:

Reviewer's Responses to Questions

**Comments to the Author**

1. If the authors have adequately addressed your comments raised in a previous round of review and you feel that this manuscript is now acceptable for publication, you may indicate that here to bypass the “Comments to the Author” section, enter your conflict of interest statement in the “Confidential to Editor” section, and submit your "Accept" recommendation.

Reviewer #1: All comments have been addressed

Reviewer #2: All comments have been addressed

2. Is the manuscript technically sound, and do the data support the conclusions?

Reviewer #1: Yes

Reviewer #2: Yes

3. Has the statistical analysis been performed appropriately and rigorously? 

Reviewer #1: Yes

Reviewer #2: Yes

4. Have the authors made all data underlying the findings in their manuscript fully available?

Reviewer #1: (No Response)

Reviewer #2: Yes

5. Is the manuscript presented in an intelligible fashion and written in standard English?

Reviewer #1: Yes

Reviewer #2: Yes

6. Review Comments to the Author

Reviewer #1: The author has addressed all the requested comments, allowing a better understanding of the manuscript for reading by any public interested in the topic.

Reviewer #2: (No Response)

7. PLOS authors have the option to publish the peer review history of their article (what does this mean?). If published, this will include your full peer review and any attached files.

Reviewer #1: **Yes: **Fátima Méndez-López

Reviewer #2: **Yes: **Alejandra Aguilar-Latorre
